# Potential Health Benefits of Ropy Exopolysaccharides Produced by *Lactobacillus plantarum*

**DOI:** 10.3390/molecules25143293

**Published:** 2020-07-20

**Authors:** Tülin Yılmaz, Ömer Şimşek

**Affiliations:** Department of Food Engineering, University of Pamukkale, 20070 Denizli, Turkey; ylmzztulin@gmail.com

**Keywords:** lactic acid bacteria, exopolysaccharide, health-benefits, antioxidant, prebiotic, immunity

## Abstract

The ability of *Lactobacillus plantarum* to produce exopolysaccharides (EPS) of various structures and properties is effective in showing both starter and probiotic culture qualification. In this study, the potential health promoting functions of the ropy EPS produced by *Lactobacillus plantarum* strains isolated from tarhana were tested. A stimulation of the pro-inflammatory IL-12 and TNF-α cytokines was observed in the presence of the ropy EPS suggesting an in vitro immune modulation. Similarly, the tested EPS demonstrated promoted the growth of the probiotic strains in fermentation medium. A medium level of radical scavenging activities of ropy EPS was observed whereas the superoxide and hydroxyl scavenging activities were more effective. The ropy EPS also showed α-glucosidase inhibition and cholesterol removal characteristics depending on their concentration. These findings revealed the potential health-promoting functions of ropy EPS from *L. plantarum* strains and EPS from *L. plantarum* PFC311 and PFC310 strains demonstrated multiple health-improving effects that can be further evaluated in food and other industries.

## 1. Introduction

Lactic acid bacteria is a wide group consisting of gram positive, acid-tolerant, industrial and health-promoting species. *Lactobacillus plantarum* is a facultative heterofermentative lactic acid bacteria (LAB) species that can be isolated from many ecological niches. *L. plantarum* is a member of the gastrointestinal tract and can be found in the microbiota of almost all fermented foods [1,2]. *L. plantarum* has been recommended and used as a probiotic strain due to the health benefits as well as starter culture in the production of distinct fermented food products [3]. Undoubtedly, the exopolysaccharides (EPS) produced by *L. plantarum* strains play a big role in the emergence of multi-faceted benefits [4]. EPS are the polymers produced by many LAB, containing long or branched repeating sugar and sugar derivatives, with high molecular weight and mainly produced by secreting into the cell wall or secreting out of the cell. EPS produced by LAB are structurally divided into two subclasses comprised of: homopolysaccharides (HoPS) consisting of repeating units of uniform monosaccharides and heteropolysaccharides (HePS) consisting of regular repeating units of a unit consisting of two or more monosaccharides [5]. Although the ecological role of these structures is estimated to be multifaceted, EPS might play roles in the survival and protection in different ecological niches [6,7]. *L. plantarum* is also very diverse in terms of EPS production [8,9], especially since they have the largest genome among lactic acid bacteria [10]. For these reasons, it is such a great resource for the production of new EPS that might be compatible with the solution of different health functions. Importantly, there is growing evidence that EPS produced by members of the LAB, including *L. plantarum*, have the potential to affect the host’s health, and dietary consumption of EPS can modulate, for example, immune function or beneficial bacteria levels in the gastrointestinal tract [5,7,8,11,12]. For example, in the presence of α-D glucan produced by *L. plantarum*, probiotic bacteria have grown well in artificial gastric juice [13]. In addition, EPS produced by *L. plantarum* RJF4 and *L. plantarum* KX041 have been found to have high antioxidant capacities with significant free radical scavenging activities [14,15]. In another study, *L. plantarum* NTU102 and *L. plantarum* JLAU103 strains induced cytokine production in the RAW 264.7 cell line (including IL-6, TNF-α and IL-1β) and cytokine formation by stimulating toll-like antigen inhibitor cells suggesting their activity as potential immune-modulator agents [16].

Recently, we characterized the structural and technological characteristics of EPS produced by distinct *L. plantarum* strains isolated from tarhana, a cereal-based fermented food product, and our findings revealed that distinct *L. plantarum* strains were capable of producing EPS with different sugar content, rheology and thermal properties [17]. In this study, the potential health-promoting functions of EPS from *L. plantarum* strains were investigated in terms of in vitro immune modulation characteristics, prebiotic properties, antioxidant activities, α-glucosidase inhibitor characteristics and cholesterol removal capabilities. These findings can be important for the understanding of the role of EPS in terms of their potential usage for the health promoting characteristics that might result in new practices for the food and pharmaceutical industries, and an enlightenment of the known health-improving properties of *L. plantarum* strains.

## 2. Results

### 2.1. Immune Modulation Characteristics of Ropy EPS

The induction of anti-inflammatory and pro-inflammatory cytokines by the ropy EPS produced by six L. plantarum strains was tested and our findings revealed that the pro-inflammatory effects of these EPS were higher because they stimulated further production of IL-12 and TNF-α cytokines (Figure 1a,b). On the other hand, it was realized that the anti-inflammatory effects of the same EPS were limited in HT-29 cells since they stimulated low levels of IL-4 and IL-10 production (Figure 1c,d). EPS samples from *L. plantarum* strains, PFC310E and PFC311E, were the most effective in terms of cytokine stimulation at low EPS concentration. In contrast, PFC312E and PFC313E have managed to induce all cytokines, at least in each of the two concentrations. However, it is worth noting that PFC310E and PFC311E have significant pro-inflammatory effects. The fact that the pro-inflammatory effects of the EPS used in this study were more successful indicated that these structures had a cytokine-stimulating effect on macrophage cells. This feature is very meaningful in terms of stimulating macrophages in the digestive tract. Similarly, heteropolysaccharides produced by *L. plantarum* NTU102 and *L. plantarum* JLAU103 strains have been shown to improve the phagocytic activities of RAW264.7 macrophages [16].

### 2.2. Prebiotic Properties of Ropy EPS

After 48 h of incubation, the growth of *Bifidobacterium bifidum* DSM 20,082 was mostly achieved when EPS produced from *L. plantarum* PFC310 strain (PFC310E was used as a carbon source rather than glucose). These results indicated that *B. bifidum* DSM 20082 partially utilized EPS from PFC310E, PFC311E and PFC312E as an energy source (Figure 2a). EPS of PFC311E, as the highest level of utilized EPS sample by the *Lactobacillus casei* subsp. *shirota* probiotic strain, started being used by the probiotic strain after the 6th hour, and the cell density detected in the positive control was reached after the 30th hour. *L. casei* subsp. *shirota* showed a dioxic growth curve with EPS from PFC311E. The same growth curve was followed by PFC312E as well. However, in the second logarithmic phase of the growth curve, the cell density increased at a limited level. In other EPS (PFC310E, PFC311E and PFC312E), cell density increment was detected at a limited level in the late hours of incubation (Figure 2b).

It was observed that the best used EPS by *Lactobacillus rhamnosus* GG was from strain PFC311E while the least used was from PFC308E (Figure 2c). In case where PFC311E was placed in the growth environment of *Lactobacillus acidophilus* DSM 20079, this strain showed dioxic growth, and cell density reached the same level as that of positive control at the end of the incubation period. *L. acidophilus* DSM 20,079 utilized EPS of PFC312E in a slowly manner. Therefore, it was understood that *L. acidophilus* DSM 20079 which is a probiotic strain, could partially use PFC311E and PFC312E (Figure 2d). These results showed a successful prebiotic effect for EPS from PFC311E, by promoting the growth of all tested probiotic strains in this study. This EPS was followed by EPS from PFC312E. Although in the growth environment, no rapid growth was observed in the presence of the EPS compared to glucose, the probiotic strains promoted their growth during the 48 h of fermentation. This was probably due to the delay of the hydrolysis and transfer process of the complex EPS during the growth process.

### 2.3. Antioxidant Properties of Ropy EPS

The superoxide anion scavenging activity varied depending on the EPS concentrations used (*p* < 0.05), and the highest activity was determined as 36.11% in EPS produced from the *L. plantarum* PFC310 strain (when 2 mg/mL concentration was used). The superoxide anion scavenging activity increased together (proportionally) with the increment of all EPS concentrations (Figure 3a). The superoxide anion scavenging mechanism is associated with the dissociation energy of the O-H bond.

As the number of electron withdrawal groups attached to the polysaccharide increases, the dissociation energy of the O-H bond weakens. Indeed, Huang et al. [18] associated the in vitro superoxide anion scavenging activity of a neutral polysaccharide and four acidic polysaccharides with the presence of some types of electrophilic groups, such as aldehyde or keto, that facilitate the release of hydrogen from the O-H bond for an effective activity. Depending on the EPS concentration, the hydroxyl radical scavenging activity increased (*p* < 0.05) and the highest activity (51.86%) was detected at the concentration of 2 mg/mL of the EPS obtained from *L. plantarum* PFC309. As the concentration of EPS of each strain increased, especially by doubling the amount of concentration, the hydroxyl radical scavenging activity values increased approximately 2 times (Figure 3b). The highest increase in radical scavenging activity due to EPS concentration was observed in PFC310E and PFC311E. EPS minimized the iron ion concentration in the Fenton reaction. The hydroxyl radical scavenging effects of these EPS might be due to the ability of EPS’s hydroxyl groups to donate active hydrogen. The results indicate that EPS of PFC309E, PFC310E and PFC311E have strong hydroxyl radical scavenging activities. The radical scavenging activities of the ropy EPS analyzed in this study were almost at a similar level. Radical scavenging activities of EPS increased depending on the concentration. The highest radical scavenging activity value was determined as 42.67% with the EPS obtained from *L. plantarum* PFC312 at a concentration of 2 mg/mL (Figure 3c). In comparison to other *L. plantarum* EPS, it was observed that radical scavenging activities remained at a medium level, however superoxide and hydroxyl scavenging activities of EPS from PFC310E and PFC311E were higher. It can be assumed that the antioxidant activity of polysaccharides might be related to molecular size, and smaller molecular size might be associated with stronger antioxidant capacity [19].

### 2.4. α-Glucosidase Inhibitor Activity of Ropy EPS

All EPS used in this study inhibited α-glucosidase enzyme activity at various levels. In all EPS samples, there was a significant (*p* < 0.05) increase in inhibition of the enzyme depending on the concentration (Figure 4a). The highest enzyme inhibition activity among the EPS occurred in EPS from PFC311E. Although the high molecular weight EPS [20] produced by *L. plantarum* BR2 strain showed a higher (67%) glucosidase inhibitory effect than our findings, it can be interpreted that the ropy EPS also had a remarkable effect.

### 2.5. Cholesterol Removal Feature of Ropy EPS

The tested ropy EPS demonstrated cholesterol lowering abilities, however it was remarked that there was a difference in % values of lowering cholesterol level depending on the EPS produced by different *L. plantarum* strains (Figure 4b). Among the EPS, PFC308E decreased cholesterol at the highest level with 68.75%, while the lowest cholesterol lowering activity was determined as 33.33% with the EPS produced from the PFCE311 strain (*p* < 0.05). Previously, the cholesterol lowering abilities of distinct EPS were shown to be ranged from 31% to 48.81% [20,21]. Our findings revealed that the cholesterol lowering activities of EPS from PFC310E, PFC311E, PFC312E and PFC313E were in a similar range with the literature whereas EPS from PFC308E and PFC309E demonstrated high levels of cholesterol lowering activities compared to the literature data. These findings suggest that the EPS might show different cholesterol removal effects depending on their sizes and structures.

## 3. Materials and Methods

### 3.1. Material

In this study, a total of six *L. plantarum* strains (PFC308, PFC309, PFC310, PFC311, PFC312 and PFC313), previously isolated from tarhana which were determined to be ropy EPS producers by Zehir et al. [17] were used. Strains were obtained from the Pamukkale University Food Engineering Culture Collection (PUFECC, WDCM 1019). These isolates were cultivated in MRS (Merck, Darmstadt, Germany) broth for 18 h at 30 °C. The working stocks of each strain were kept at −20 °C in their glycerol stocks (30%).

### 3.2. Ropy EPS Production and Purification

For the production of ropy EPS, *L. plantarum* strains were grown for 18 h in 2 L using an autoclavable and controllable fermenter system (Minifors, Bottmingen-Basel, Switzerland) in a modified BHI medium containing beef heart 5 g/L, calf brains 12.5 g/L, disodium hydrogen phosphate 2.5 g/L, peptone 10 g/L, sodium chloride 5 g/L, glucose 10 g/L, sucrose 20 g/L, Tween 20 g/L, sodium acetate 5 g/L, peptone casein 5 g/L, peptone meat 5 g/L and magnesium sulfate 0.2 g/L at 30 °C with stirring 100 rpm and without aeration. The fermentation medium was adjusted to pH 6.0 with the automatic system of the fermenter using 5 M NaOH and 5 M HCl. The EPS producer *L. plantarum* strains were cultivated in 20 mL MRS at 30 °C prior to inoculation of the fermenter. The isolation of ropy EPS from bacterial cultures was performed according to the method specified by Dertli et al. [22]. Firstly, cells were separated by bacterial biomass and supernatant by centrifugation for 30 min at 4 °C, 7200× *g* and the supernatant was obtained. To this supernatant, cold ethanol at equal amounts was added and the EPS was precipitated at 4 °C overnight and EPS was retained by centrifugation at 7200× *g* at 4 °C, for 30 min. The supernatant was removed and the extracted EPS was dissolved in 50 mL dH_2_O (In cases where it is difficult to dissolve, a slight heating process was done at 50 °C), then 2 times as much cold ethanol was added and kept at 4 °C for overnight. At the end of the period, the extracted EPS was centrifuged again and the supernatant was removed, after that the pellet was dissolved in 30 mL of dH_2_O. To this EPS solution, trichloroacetic acid (TCA) was added until a final concentration of 15% was reached which was followed with 4 h of incubation at 4 °C with gentle shaking. This solution was centrifuged at 4 °C, 7200*×* g for 30 min to separate impurities from proteins in the EPS samples. After this process, the supernatant was taken and after adjusting its pH to 7.0 with 5 M NaOH, 5 times as much cold ethanol was added. Afterwards, EPS were obtained and subjected to lyophilization to obtain the lyophilized EPS.

### 3.3. Determination of the Immune Modulation Properties of Ropy EPS

Lyophilized ropy EPS produced by *L. plantarum* PFC308, PFC309, PFC310, PFC311, PFC312 and PFC313 strains were dissolved using DMEM at a concentration of 1 mg/mL. HT-29 cells were prepared by pre-passaging and were added to 6-well cell culture dishes in such a way that 400,000 cells were presented per well, and incubated for 24 h in 3 mL DMEM medium. After the incubation period, each of the ropy EPS mentioned above was applied to the cells attached to the base of the cell culture dishes at 50 and 100 μg/mL con. and the cells were left to incubate for 24 h. Only DMEM medium was added to the control group. After the 24-h incubation period, the mediums found in the 6-well plates were collected into 15 mL centrifuge tubes and IL-4, IL-10, IL-12 and TNF-α cytokine concentrations were determined using ELISA kits (Sunlong Biotech, Hangzhou, China).

### 3.4. Determination of Prebiotic Properties of Ropy EPS

The prebiotic properties of ropy EPS were determined by examining their effects on the growth of probiotic strains [23]. For this purpose, *Bifidobacterium bifidum* DSM 20082, *Lactobacillus acidophilus* DSM 20079, *Lactobacillus rhamnosus* GG and *Lactobacillus casei* subsp. *shirota* were used. Bifidobacteria and Lactobacilli were grown at 37 °C under anaerobic conditions in MRS broth medium for 24 h. To determine the prebiotic effect, sugar-free MRS (pH 5.7) was prepared and 0.5% EPS was used as carbon source. MRS medium containing 0.5% glucose was used as positive control while sugar-free MRS medium was used as negative control. Probiotic strains were inoculated as 10^6^ CFU/mL from the previous culture to EPS containing MRS medium and the cell density alteration was detected in a multi-plate reader (Multiscan, Thermo Fisher Scientific, Waltham, MA, USA) at 600 nm for 48 h once every 15 min.

### 3.5. Determination of Antioxidant Activities of Ropy EPS

The determination of superoxide anion scavenging activity of EPS was performed according to the method specified by Li et al. [24]. Briefly, 1 mL of phosphate buffer (50 mM, pH 8.34) and EPS at different concentrations of 0.5, 1.0 and 2.0 mg/mL were incubated by mixing for 20 min at 25 °C. Subsequently, by adding 0.2 mL pyrogallol (3 mM), the optical density of the mixture was measured for 5 min at 325 nm once every 10 s. The superoxide scavenging activity (%) was calculated by using the equation given below:Superoxide scavenging (%) = [(ΔA_0_ − ΔA_1_)/ΔA_0_] × 100(1)
where ΔA_0_ is the absorbance difference every 10 s in solutions without sample and ΔA_1_ is the absorbance difference of the solutions of different concentrations in 10 s.

The hydroxyl radical scavenging effect of EPS was determined by Fenton reactions. Accordingly, 1 mL brilliant green (0.435 mM), 2 mL FeSO_4_ (0.5 mM), 1.5 mL H_2_O_2_ (3% *w*/*v*) and the mixture containing EPS at different concentrations of 0.5, 1.0 and 2.0 mg/mL were incubated for 1 h at 37 °C. Then, the optical density was measured at 624 nm by centrifugation at 4000× *g* for 5 min. The hydroxyl radical scavenging activity of EPS was performed by using the equation given below:OH scavenging (%) = [(A_0_ − A_1_)/(A − A_1_)] × 100(2)
where A_0_ is the solution containing samples in different concentrations, A_1_ is the sample-free solution and A is the solution containing Fenton reactions but without sample.

Radical scavenging activity of EPS was determined by using the method specified by Zhang et al. [25]. Accordingly, ropy EPS in different concentrations (0.5, 1.0 and 2.0 mg/mL) were added to 2 mL ethanolic DPPH radical solution (0.2 mM). The mixture was vigorously stirred and incubated for 1 h in a dark room at room temperature. Then, the optical density of the supernatant obtained as a result of 10 min centrifuge (Hettich Universal 30 RF Vestfalya, Germany) at 8000× *g* was measured at 517 nm (Multiscan Go, Thermo Fisher Scientific, Waltham, MA, USA). The radical scavenging activity of EPS was performed by using the equation given below:Radical scavenging (%) = [(A_0_ − A_1_)/A_0_] × 100(3)
where A_0_ is the DPPH solution without sample and A_1_ refers to the solution containing samples in different concentrations.

### 3.6. Determination of α-Glucosidase Inhibitor Activities of Ropy EPS

α-glucosidase inhibitory activities of ropy EPS were determined by using the method specified by Kazeem et al. [26]. For this, 100 µL α-glucosidase (1.0 U/mL, from *Saccharomyces cerevisiae*, Sigma, St. Louis, MO, USA) and different concentrations of 50 µL EPS (0.5, 1.0 and 2.0 mg/mL) were incubated for 10 min at 37 °C. Then, to start the reaction, 50 µL of 3.0 mM 4-nitrophenly α-D-glucopyranoside (pNPG) dissolved in 20 mM phosphate buffer was added and incubated for 20 min at 37 °C, then 2 mL of 0.1 M Na_2_CO_3_ was added and the reaction was stopped. Yellow paranitrophenol released from pNPG was measured at 405 nm (Multiscan Go, Thermo Fisher Scientific, Waltham, MA, USA). The α-glucosidase inhibitor activity (%) was calculated by using the equation given below:α-glucosidase inhibitor activity (%) = [(A_Control_ − As_ample_)/A_control_] × 100(4)
where A_Control_ is the solution without sample and A_Sample_ is the solution containing samples of different concentrations.

### 3.7. Determination of Cholesterol Removal Capabilities of Ropy EPS

Ropy EPS’s cholesterol-removing capability was determined by using the method specified by Soh et al. [27]. After preparing 1 mL reaction mixture containing 0.1% ropy EPS and 30 µg cholesterol, the reaction mixture was incubated for 20 min at 25 °C and 50 µL hexadecyl trimethyl ammonium bromide was added to the mixture. The mixture was then centrifuged at 12,500× *g* and the optical density of the supernatant was measured at 500 nm. The cholesterol-removing capability was calculated by using the equation given below:Cholesterol-removing capability (%) = [(A_Control_ − A_Sample_)/A_Control_] × 100(5)
where A_Control_ is the cholesterol solution without EPS and A_Sample_ is the cholesterol solution containing 0.1% EPS.

### 3.8. Statistical Analysis

In the study, all the analyses were replicated at least three times and the statistical analysis on the effects of EPS obtained from *L. plantarum* strains on the abovementioned activities were analyzed in terms of the differences between concentrations by using the one-way ANOVA test in Minitab 16.0 Statistical Software package program (State College, PA, USA). Tukey’s test was used to compare the difference between the samples (*p* < 0.05).

## 4. Conclusions

All these results showed that ropy EPS produced by *L. plantarum* strains have some health-improving effects. It has also been revealed that some of these EPS have multiple health-improving effects. EPS from PFC310E and PFC311E stand out in particular as important EPS in terms of protecting and improving health due to their prebiotic, antioxidant, effect on cholesterol level, their success in α-glucosidase enzyme inhibitor activity and their ability to induce immune cytokines in the HT-29 cell line.

## Figures and Tables

**Figure 1 molecules-25-03293-f001:**
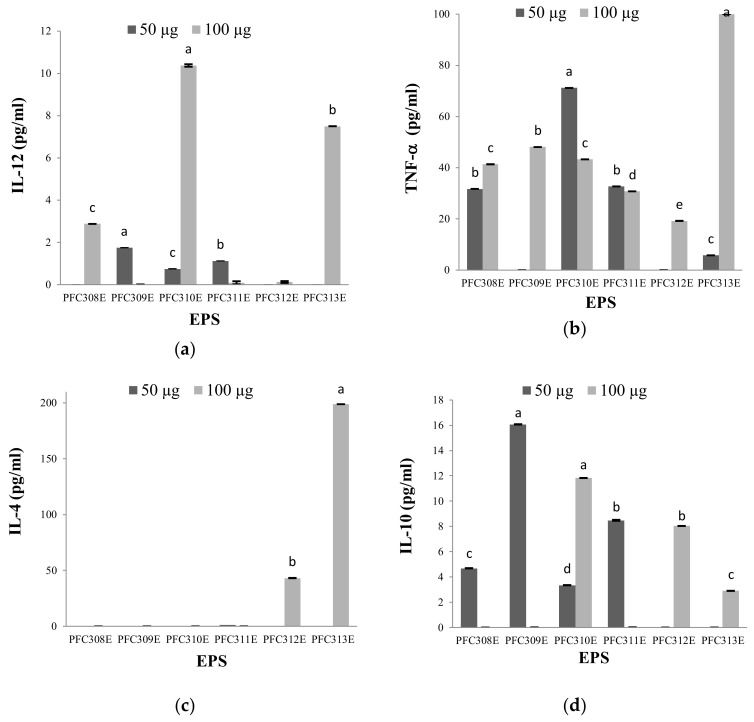
The cytokine response concentrations (pg/mL) at HT-29 cell lines stimulated by ropy EPS produced from *L. plantarum* strains. Cytokines (**a**) IL-12, (**b**) TNF-α, (**c**) IL-4 and (**d**) IL-10. Different letters show differences (*p* < 0.05) between the EPS at each cytokine concentration.

**Figure 2 molecules-25-03293-f002:**
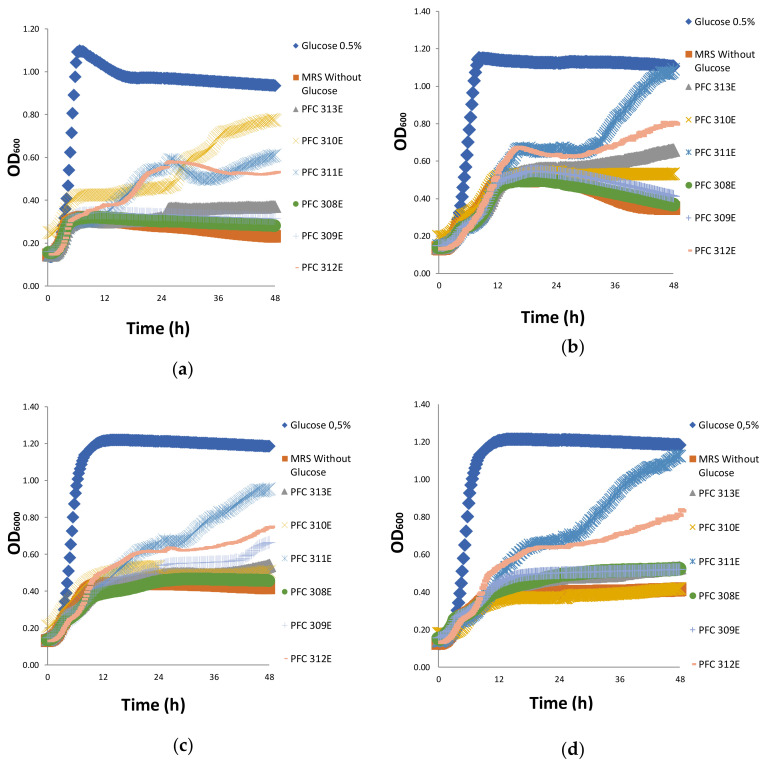
The growth curve of probiotic strains (**a**) *B. bifidum* DSM 20082, (**b**) *L. casei* subsp. *shirota,* (**c**) *L. rhamnosus* GG and (**d**) *L. acidophilus* DSM 20079 with EPS and glucose.

**Figure 3 molecules-25-03293-f003:**
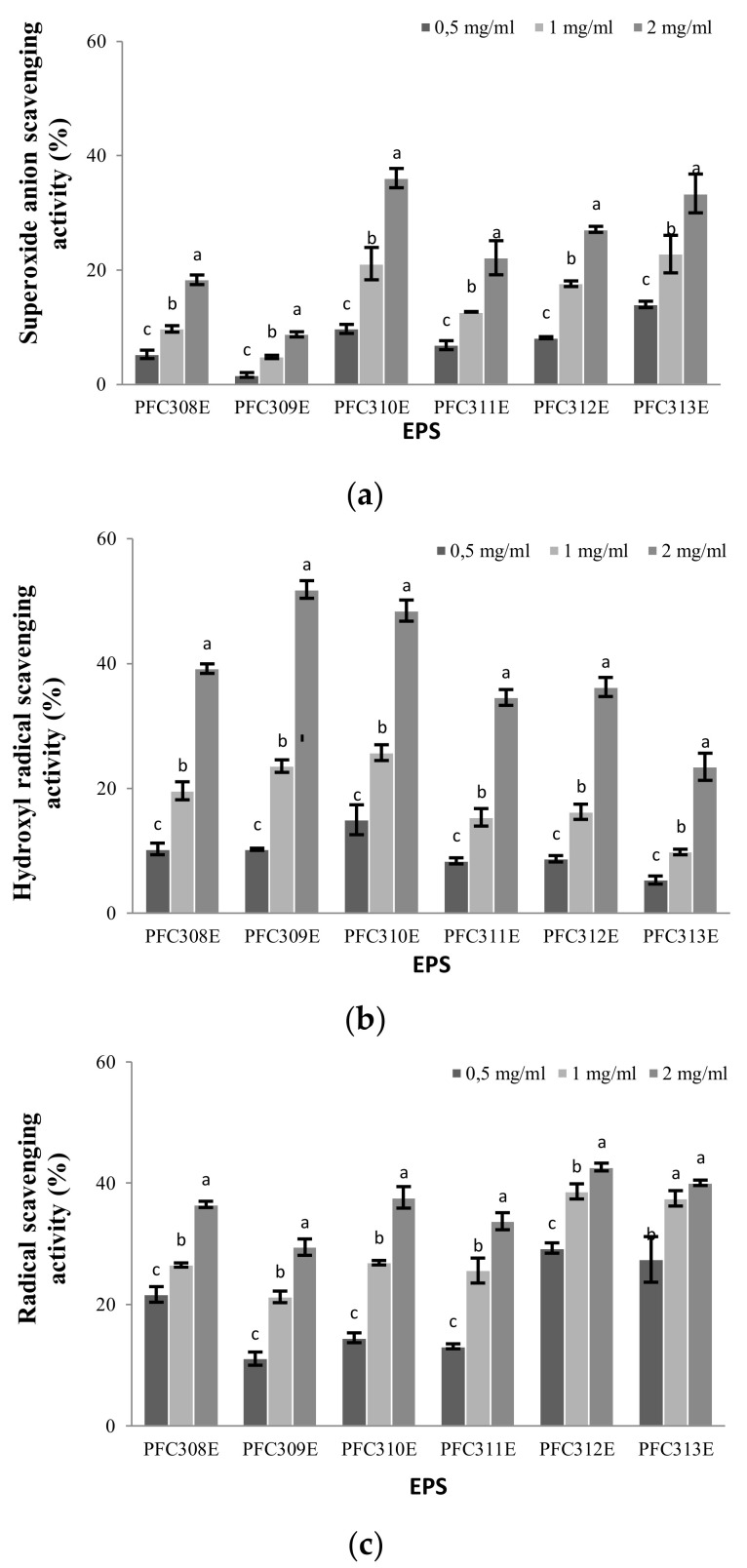
The antioxidant activity of ropy EPS at different concentrations. Scavenging activities (**a**) Superoxide anion, (**b**) hydroxyl radical and (**c**) radical. Different letters show the differences (*p* < 0.05) between the concentrations at each EPS.

**Figure 4 molecules-25-03293-f004:**
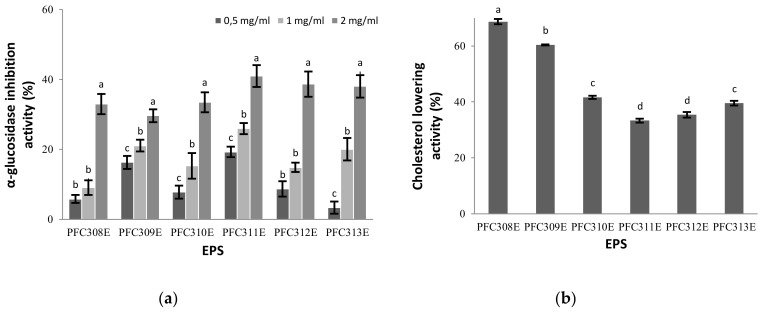
The (**a**) α-glucosidase inhibitory and (**b**) cholesterol lowering activities of ropy EPS at different concentrations. Different letters show the differences (*p* < 0.05) between the concentrations at each EPS at (**a**) and EPS at (**b**).

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
