# Peer review of "Potential Health Benefits of Ropy Exopolysaccharides Produced by Lactobacillus plantarum"

_molecules, 2020, doi:10.3390/molecules25143293_

Round 1

Reviewer 1 Report

Manuscript: molecules 833927

Title: Potential health benefits of ropy exopolysaccharides produced by Lactobacillus plantarum

Tülin Yılmaz and Ömer Şimşek evaluated the ability of Lactobacillus plantarum to produce exopolysaccharides (EPS) of various structures and properties. In their study, the potential health promoting functions of the ropy EPS produced by Lactobacillus plantarum strains isolated from tarhana were tested. Their results revealed the potential health-promoting functions of ropy EPSs from L. plantarum strains.

Because of this, the current study is on a topic of relevance and general interest to the readers of the journal.

The design of the study seems quite useful for the purpose and the manuscript displays interesting information. However some modifications must be carried out before a possible consideration.

In several points, the discussion of the obtained results are not in line with the presented results or absent. Authors should discuss the results and how they can be interpreted in perspective of previous studies and of the working hypotheses. The findings and their implications should be discussed in the broadest context possible and limitations of the work highlighted. Future research directions may also be mentioned.

I found the paper not to be well written. I recommend that you have your manuscript professionally edited before submission or read by a native English-speaking colleague.

See below a point by point:

Please, check in all the text the use of EPSs or EPS abbreviation.

L25. check “heterofermantative”

L25. Please, It would be of interest to the reader to have some information on the lactic acid bacteria (LABs).

L27. Please, use “microbiota” instead of “microflora”.

L39. Please, check this part: “which can be originated from its genome as the largest genome”.

L42. Check “LAB family”, is not correct.

L62. I think that this section was combined with the Discussion section.

L63. Please, add more explanation and/or references about the immune modulation characteristics of ropy EPSs. The discussion of this very important part is inadequate.

L64. Based on the hypothesis that the obtained findings can be important for the understanding of the role of EPSs in terms of their potential usage for the health promoting characteristics that might result in new practices for the food industries, can you please provide the initial and final amount of EPS before and after lyophilization?

L65. L. plantarum and the others bacterial name should be in italics.

L79. When EPS produced from L. plantarum PFC310 strain (PFC310E)...

L79. Please, what do you mean with “achieved”? Check this sense because it doesn't agree with Figure 2. How did the authors conclude about the effect of ropy EPSs on probiotic cultures? Can you add some details about the choose of the above mentioned probiotic cultures?

L121-123. “The hydroxyl radical scavenging effects of these EPSs might be due to the ability of EPS's hydroxyl groups to donate active hydrogen”. Can you please provide some explanation and/or references about this?

L130-132. “It can be assumed that the antioxidant activity of polysaccharides might be related to molecular size, and smaller molecular size might be associated with stronger antioxidant capacity [20]”. Which molecular size? No molecular size were measured in this work.

L104-132. Authors should discuss the results.

L144. “depending on the strains”. Can you please provide some explanation and/or references about this?

In general, the authors should indicate if the experiments were carried out with or without replicates.

L164. Please, add more details about the preparation of the bacterial inoculum.

L165. It would be of interest to the reader to have more detailed information on the fermenter system.

L171. Dertli et al. [23] there isn’t in the reference list. Please check.

L178. Which was the initial concentration of trichloroacetic acid (TCA)?

L200. Please, Indicate the medium and the conditions during probiotic strains inoculum revival and preparation and finally indicate the amount of inoculum added in the test medium and the initial cell density. The same holds for positive control medium. Provide some information on the preparation of the medium used for the prebiotic assay. Here and after, it would be of interest to the reader to have more detailed information on the choose of the level of added ropy EPS in the different assays.

L202. pH was not determined?

Figure 1. The legend should indicate a concentration. Which is the role of the different letters on the bars?

Figure 2. Please, in the text you referred to the time using hours while on the x axis “minutes”. Please, let’s uniform the parameter.  I suggest that this figure is depicted as bars and not lines. It would be easier to understand the difference among the different growth procedures.

Figure 3. Which is the role of the different letters on the bars?

Figure 4. Which is the role of the different letters on the bars?

Check all the manuscript for mistakes in the references.

L230. Please, check and correct Kazeem et al. [27] and Soh et al. [28].

Please, check the references 15 and 17.

Author Response

Thank you for fruitful comments which give chance to improve our manuscript. We did our best and try to obey the comments raised.

Best regards.

Reviewer 2 Report

The study “Potential health benefits of ropy exopolysaccharides 2 produced by Lactobacillus Plantarum” is well designed and performed. There are still some questions that need to be stated or modified.

Comments for the authors:

  1. Specific the main function of the ropy EPSs produced by L. Plantarum strains in the title to make it clear.
  2. In Materials and Methods, please add the HT-29 cell line information and cell culture information.
  3. Figure 1c, it’s better to cut off the y-axis for PFC313E in order to show up the former four groups. The authors could try 1a and 1d to see if they can get better figures.
  4. Figure 2, some of the colors are very close and hard to be differentiated from each other. I couldn’t find PFC 310E as purple color in the figures. No way to understand if the authors’ results are good. Please replace the similar color, for instance, black can be chosen as a replacement.
  5. Line 103-132, did the authors perform the statistics among different strains under the same doses? It’s better to get a clear mind in which strain is beneficial to which function or if they are universally good to the function.
  6. Figure 3, revise the maximum value in the y-axis to an appropriate one, like 60, then the column chart ratio will be better. Figure 4 as well.
  7. Try to update the literature to the most recent ones.

Author Response

(The authors gave the same response as above.)

Round 2

Reviewer 1 Report

For my opinion, the manuscript can be accept in present form